# Enhancing Cross-Lingual Entity Alignment in Knowledge Graphs through Structure Similarity Rearrangement

**DOI:** 10.3390/s23167096

**Published:** 2023-08-10

**Authors:** Guiyang Liu, Canghong Jin, Longxiang Shi, Cheng Yang, Jiangbing Shuai, Jing Ying

**Affiliations:** 1School of Computer and Computing Science, Hangzhou City University, Hangzhou 310015, China; zju_liu@zju.edu.cn (G.L.); shilx@hzcu.edu.cn (L.S.);; 2College of Computer Science and Technology, Zhejiang University, Hangzhou 310027, China; 3Zhejiang Academy of Science & Technology for Inspection & Quarantine, Hangzhou 310051, China; sjb@zaiq.org.cn

**Keywords:** knowledge graph, cross-lingual entity alignment, structural similarity rearrangement

## Abstract

Cross-lingual entity alignment in knowledge graphs is a crucial task in knowledge fusion. This task involves learning low-dimensional embeddings for nodes in different knowledge graphs and identifying equivalent entities across them by measuring the distances between their representation vectors. Existing alignment models use neural network modules and the nearest neighbors algorithm to find suitable entity pairs. However, these models often ignore the importance of local structural features of entities during the alignment stage, which may lead to reduced matching accuracy. Specifically, nodes that are poorly represented may not benefit from their surrounding context. In this article, we propose a novel alignment model called SSR, which leverages the node embedding algorithm in graphs to select candidate entities and then rearranges them by local structural similarity in the source and target knowledge graphs. Our approach improves the performance of existing approaches and is compatible with them. We demonstrate the effectiveness of our approach on the DBP15k dataset, showing that it outperforms existing methods while requiring less time.

## 1. Introduction

Recently, knowledge graphs (KGs) have gained widespread adoption in AI-related fields, such as decision systems, knowledge reasoning, and recommendation systems, due to their excellent performance in storing structured data [1]. Knowledge graphs commonly consist of knowledge represented in the form of a triple, denoted by (h,r,t), in which *h* and *t* are entities, and there is a relation *r* between them. Although effective in representing structured data, problems caused by the underlying symbolic nature of such triples such as a lack of explicit semantics, absence of complex operations, and expressivity limitations usually make KGs hard to manipulate. To address these issues, research efforts have focused on the distributed representation of KGs, also known as KG embedding. KG embedding models embed a KG into a low-dimensional vector space by representing its entities and relations as semantic information vectors. The earliest work based on translation, TransE [2], considers a relation as the translation from its head entity to its tail entity. TransE can learn embeddings for all the elements of a ground truth triple (h,r,t) by assuming h+r≈t, where h, r, and t are the embeddings of *h*, *r*, and *t*, respectively. Subsequent works have focused on enhancing TransE, such as TransD [3], TransH [4], and PTransE [5].

However, constructing KGs usually involves independent organizations or individuals, resulting in incomplete KGs that may not meet the needs of some applications. The above methods mainly model a single KG, but to obtain more comprehensive knowledge, it is necessary to integrate different knowledge graphs. Entity alignment [6] plays a critical role in knowledge fusion tasks. Entity alignment models fuse KGs by aligning pairs of entities representing the same concept in the real world from different KGs, which may be in different languages. Cross-lingual entity alignment [7] integrates two or more KGs from other languages via pre-aligned seeds. Traditional entity alignment techniques focus on two perspectives: one is based on the equivalence reasoning specified by OWL semantics [8], and the other is based on similarity computation, which compares the symbolic features of entities [9,10]. Some studies have also focused on improving the accuracy of entity alignment through machine learning [11,12] and crowdsourcing [13].

Recent works devote a lot of effort to machine learning and deep learning approaches for knowledge graph entity alignment. MTransE [14] embeds KGs from different languages into different vector space, and learns a matrix for spatial transformation between monolingual vector spaces. JAPE [15] jointly embeds both the structural and attribute information of knowledge graphs into a unified vector space. BootEA [16] proposes a bootstrapping approach to iteratively generate new aligned entities as training data. In particular, deep learning methods have shown promising results in this area. GCN-Align [17] trains GCNs with pre-aligned seeds to embed entities of each language into a unified vector space. HGCN-JE [6] jointly learns entity and relation representations. RDGCN [18] incorporates relation information via attentive interactions between the knowledge graph and its dual relation counterpart to capture neighboring structures and learn better entity representations.

We find that all the methods mentioned above suffer from a limitation in that they only consider the globally closest entity in the vector space as an equivalent entity, which can be seen as a global optimum. However, when considering a specific node, the entity selected based on distance alone may not be the optimal cross-lingual counterpart, as it does not take into account the structural similarity between nodes. We can draw inspiration from SEU [19], which transformed cross-lingual knowledge graph entity alignment into an assignment task, as the original knowledge graph and the target knowledge graph share a similar graph structure, i.e., their adjacency matrices are similar. In this paper, we propose a cross-lingual entity alignment method based on local structural rearrangement. Our method calculates a k-nearest neighbor candidate set for each entity of the original knowledge graph during the alignment phase of the model. Subsequently, we rearrange each of the obtained candidate sets based on local structure, consisting of the surrounding nodes, to enhance the matching accuracy of equivalent entities.

As illustrated in Figure 1, the source knowledge graph (Chinese) contains relations between entities such as **“Ping Guo”**($N_{s_{1}}$) and **“Qiao Bu Si”**($N_{s_{2}}$), **“Jia Zhou”**($N_{s_{3}}$) and **“Wei Ruan”**($N_{s_{5}}$). For instance, **“Qiao Bu Si”** founded **“Ping Guo”** and **“Ping Guo”** is located in **“Jia Zhou”**. When searching for the cross-lingual correspondence of entity **“Ping Guo”** in the target knowledge graph (English), there are two possible results: **“Apple”**($N_{t_{a}}$) and **“Apple”**($N_{t_{b}}$). In this case, the local structure of entities must be taken into consideration. As the cross-lingual equivalent entities include **“Steven Jobs’’**($N_{t_{2}}$) in the target KG and **“Qiao Bu Si”** in the source KG, **“California”**($N_{t_{3}}$) in the target KG and **“Jia Zhou”** in the source KG, and **“Microsoft”**($N_{t_{5}}$) in the target KG and **“Wei Ruan”** in the source KG, entity **“Apple”**($N_{t_{a}}$) has a similar structure to entity **“Ping Guo”**. Therefore, entity **“Apple”**($N_{t_{a}}$) is more likely to be the cross-lingual counterpart of **“Ping Guo”** than the other entity **“Apple”**($N_{t_{b}}$). In summary, we summarize the three significant contributions as follows:We introduce an approach called structure similarity rearrangement (SSR) to improve the precision of cross-lingual knowledge graph entity alignment, which centers around the formulation of a joint evaluation function that incorporates the local structural similarity of entities.We perform extensive experiments on the publicly available datasets DBP15k ZH_EN, JA_EN, and FR_EN, and demonstrate that our proposed model outperforms other state-of-the-art methods in the evaluation metrics of Hits@1.By conducting comparative experiments, we have discovered that our proposed method exhibits a seamless integration capability with other alignment models, leading to a substantial improvement over techniques (such as GCNs) that solely focus on node-level features. Moreover, our model demonstrates considerable accuracy gains when compared to approaches (e.g., RDGCN) that consider the surrounding context.

## 2. Related Work

The task of knowledge graph entity alignment can be broken down into two primary stages: (i) knowledge graph representation, and (ii) entity alignment based on the entity-level representation. In recent years, knowledge graph embedding techniques have been widely employed for distributed representation of the structural information in knowledge graphs.

### 2.1. Knowledge Graph Embedding

Knowledge graph (KG) is a technique that uses graph models to describe knowledge and model the association relationships between things [20,21,22]. KGs are composed of triples, 〈entity,relation,entity〉, and entities that have attribute–value pairs, which are connected by relationships to form a web-like structure [23,24]. However, manipulating KGs can be challenging due to the symbolic nature of triples. To address this issue, knowledge graph embedding (KGE) [25] has emerged as a promising research direction. KGE aims to map KG components into a continuous low-dimensional vector space that preserves the original structure of the KG and simplifies operations. Various embedding methods have been proposed, including TransE [2], which interprets a relation as the translation vector from the head entity to the tail entity in embedding space; TransH [4], which models a relation as a vector on a specific relationship hyperplane and learns different representations for an entity; and TransD [3], which uses two vectors to represent the semantics of an entity (relation) and construct a mapping matrix dynamically. Recently, a GCN-based approach called CompGCN [26] has been proposed, which leverages a range of entity–relation composition operations from KG embedding techniques to jointly embed both nodes and relations in a multi-relational graph.

### 2.2. Entity Alignment

Entity alignment is a crucial task in knowledge graph (KG) research which aims to identify entities that correspond to the same real-world object across different KGs. In the past, manual feature extraction or crowdsourcing [10,27,28,29] were often used to accomplish this task, which required substantial manual involvement. Recently, KG embedding techniques have been widely employed to facilitate entity alignment. JE [30] proposed to jointly learn embeddings of multiple KGs in a uniform vector space to align entities in different KGs. MTransE [14] embeds two KGs into different vector spaces and aligns them by learning two transforming matrices. GCN-Align [31] is the first to employ graph convolutional networks (GCNs) to encode entities and attributes into a unified space for entity alignment.

## 3. Problem Formulation

### 3.1. Knowledge Graph

A knowledge graph KG is composed of sets of entities *E*, relations *R*, and triples *T*, where T⊆E×R×E and KG=(E,R,T).

### 3.2. Knowledge Graph Entity Alignment

An entity alignment model aims to automatically identify all of the corresponding entities of two given KGs. Without loss of generality, we choose one KG as the source knowledge graph and the other as the target knowledge graph, and denote them as $KG_{s}=\left(E_{s},R_{s},T_{s}\right)$ and $KG_{t}=\left(E_{t},R_{t},T_{t}\right)$.

**Candidate set and ranking indicator**. In the process of entity alignment, the model endeavors to discover a fixed number of cross-lingual counterparts in $KG_{t}$ for each entity node $e_{i}$ originating from $KG_{s}$. Subsequently, these counterparts are sorted in ascending order based on vector distance, yielding the candidate set $CAND(e_{i})$ specific to the respective entity node. For any node $e_{j}$ in the candidate set of $e_{i}$ from $KG_{t}$, its ranking indicator based on vector distance can be defined as the difference between the length of the candidate set (cand) and its position in the candidate set ($index(CAND\left(e_{i}\right),e_{j})$).

**Similarity of local structures**. Let *G* represent a graph, which can be denoted as G=〈V,E〉, where *V* and *E* correspond to the sets of nodes and edges in the graph, respectively. We refer to the graph structure of $KG_{s}$ as $G_{s}$, and similarly, the graph structure of $KG_{t}$ as $G_{t}$. The local structure of a vertex is defined by the edges directly adjacent to it and the vertices connected to it via those edges. Thus, for a given node $v_{s}$ from $G_{s}$ and a node $v_{t}$ from $G_{t}$, we employ the symbol S to represent the degree of similarity between their respective local structures.

**Problem Statement** Given two knowledge graphs $KG_{s}$ and $KG_{t}$, the crux of the problem is to compute and rearrange the candidate set associated with each entity originating from $KG_{s}$.

The symbols used in our method are summarized in Table 1.

## 4. The Proposed Approach

Most of the existing models for entity alignment in knowledge graphs are composed of two main modules: an embedding module and an alignment module. The former is responsible for generating distributed representations of the input knowledge graphs, whereas the latter aims to identify similar cross-lingual counterparts in the vector space. However, the aforementioned methodologies primarily focus on devising intricate network models to acquire a highly precise distributed representation of the knowledge graph. Nevertheless, the alignment phase still adheres to the conventional approach of employing a simple greedy search. Taking inspiration from SEU [19], we acknowledge that when it comes to two knowledge graphs that pertain to the same domain, their overall graph structure exhibits similarities, and the structure surrounding equivalent entities is also comparable. In this work, we propose a structure-based approach for rearranging the candidate sets obtained by the alignment module. Specifically, given two knowledge graphs $KG_{s}$ and $KG_{t}$ in different languages, our model first generates their distributed representations via the embedding module. Next, the alignment module employs the KNN algorithm to search for cross-lingual counterparts in the target knowledge graph for each entity in the source knowledge graph, and generates the corresponding candidate sets. We assume that the graph structures around the alignment nodes in the source and target knowledge graphs are similar, such that entities with similar graph structures in the source knowledge graph are likely to have similar counterparts in the target knowledge graph.

In order to enhance the precision of cross-lingual entity alignment, we propose a novel approach to rearrange the candidate sets of nodes in consideration of the similarity between the graph structures of the nodes in the source KG and their candidate set in the target KG. This structure-based rearrangement aims to modify the distribution of the candidate sets, and, therefore, improve the overall accuracy of the alignment task.

### 4.1. Overall Architecture

As illustrated in Figure 2, we propose a straightforward approach to enhance the precision of cross-lingual entity alignment. Firstly, the embedding representation of the knowledge graph is generated by leveraging an embedding module, such as JAPE, GCNs, HGCN, or RDGCN. Subsequently, a KNN algorithm is typically employed as the alignment module to obtain the cross-lingual corresponding candidate set for each node in the source knowledge graph. To optimize this candidate set, a rearrangement algorithm is utilized. The detailed process of our proposed approach is elaborated in Algorithm 1.
**Algorithm 1:** Structural Enhancement Rearrangement Algorithm. **Input**: Entity Embeddings vec, Test Data (Es,Et), Size of candidate set cand, Gs=〈Vs,Es〉, Gt=〈Vt,Et〉 **Output**: Rearranged Candidate Sets CANDnew, Hits@1.
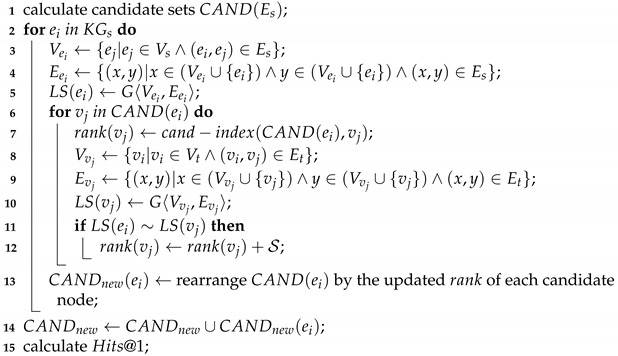



### 4.2. Embedding Module

The proposed approach primarily focuses on the structure-based rearrangement of candidate sets, and, thus, the choice of embedding module is flexible. For the purpose of illustration, we adopt the GCN alignment model as the embedding module. GCNs [17] are neural networks that can directly process graph-structured data of arbitrary shape and size. Specifically, a GCN layer receives both the node feature vector and the structural information of a graph. The structural information of the graph can be learned, and the feature vectors of all nodes can be updated through a GCN layer. By aggregating information from the neighboring nodes, GCNs update the feature vectors of nodes and are commonly employed in graph classification and regression tasks.

The input to a GCN model layer is the vertex feature matrix of the graph and the output is a new feature matrix obtained using the following equation:(1)$$H^{\left(l+1\right)}=\sigma ({\hat{D}}^{-\frac{1}{2}}\hat{A}{\hat{D}}^{-\frac{1}{2}}H^{\left(l\right)}W^{\left(l\right)}).$$
where $H^{\left(l\right)}\in R^{n\times d^{\left(l\right)}}$ is a matrix of $d^{\left(l\right)}$-dimensional feature vectors of *n* nodes in the *l*-th layer; σ is an activation function; $A\in R^{n\times n}$ is the connectivity matrix of the graph; $\hat{A}=A+I$, and $I\in R^{n\times n}$ is the identity matrix; $\hat{D}\in R^{n\times n}$ is the diagonal node degree matrix of $\hat{A}$; and $W^{\left(l\right)}\in R^{d^{\left(l\right)}\times d^{\left(l+1\right)}}$ is the weight matrix of the *l*-th layer of the GCN model; the new vertex features have $d^{\left(l+1\right)}$ dimensions.

The GCN-Align model integrates the structural and attribute information of nodes to generate node embeddings, and since our method rearranges nodes in a candidate set based on structural similarity, the attribute information of nodes is not considered and only structural information is utilized for training.

### 4.3. Calculate Candidate Sets

A cross-lingual entity counterpart is predicted by computing the distance between the entity embeddings of two KGs in the vector space. For each pair of nodes, $e_{i}$ in $KG_{1}$ and $v_{j}$ in $KG_{2}$, the distance between them is computed using the following equation:(2)$$D\left(e_{i},v_{j}\right)=\left|\right|e_{i}-v_{j}{\left|\right|}_{1}.$$
where $e_{i}$ and $v_{j}$ denote the embeddings of $e_{i}$ and $v_{j}$, respectively.

For an entity $e_{i}$ in $KG_{1}$, our model computes the distance between $e_{i}$ and all entities in $KG_{2}$ and returns a list as the candidate set in which the entities are sorted by distance from smallest to largest, because two entities with smaller distance are more likely to be equivalent. As a result, for *n* entity nodes in $KG_{1}$, a candidate sets matrix $M_{c}\in R^{n\times l_{c}}$ can be obtained, where $l_{c}$ denotes the length of each candidate set.

### 4.4. Local Structural Similarity

In the graph theory domain, the local structural analysis [32] of a node $e_{i}$ in a graph G=〈V,E〉 has been widely studied. Herein, *V* and *E* correspond to the sets of nodes and edges in the graph, respectively. Specifically, the local structure LS of the node $e_{i}$ is formally defined as follows:(3)$$LS\left(e_{i}\right)=G\langle V_{e_{i}},E_{e_{i}}\rangle .$$
where $V_{e_{i}}=\left\{e_{j}|e_{j}\in V\wedge \left(e_{i},e_{j}\right)\in E\right\}$ and $E_{e_{i}}=\left\{\left(x,y\right)|x\in \left(V_{e_{i}}\cup \left\{e_{i}\right\}\right)\wedge y\in \left(V_{e_{i}}\cup \left\{e_{i}\right\}\right)\wedge \left(x,y\right)\in E\right\}$. The local structure of each node is a subgraph of the whole knowledge graph and there are a lot of methods [33] to determine whether two graphs are similar. We propose a simple but novel equation for computing graph similarity in this article, which is outlined as follows:(4)$$S_{node}\left(e_{i},v_{j}\right)=\sum_{x\in V_{e_{i}}}[\epsilon -\min_{y\in V_{v_{j}}}D\left(x,y\right){]}_{+}$$
(5)$$S_{edge}\left(e_{i},v_{j}\right)=\sum_{\left(x_{i},y_{i}\right)\in E_{e_{i}}}\left|\left\{\left(x_{j},y_{j}\right)|cond\right\}\right|.$$
(6)$$S\left(LS\left(e_{i}\right),LS\left(v_{j}\right)\right)=\lambda \times S_{node}\left(e_{i},v_{j}\right)+\delta \times S_{edge}\left(e_{i},v_{j}\right).$$
where cond in Equation (Equation 5) denotes $\left(x_{j},y_{j}\right)\in E_{v_{j}}\wedge x_{j}\in \mathrm{CAND}\left(x_{i}\right)\wedge y_{j}\in \mathrm{CAND}\left(y_{i}\right)$; $S_{node}\left(e_{i},v_{j}\right)$ in Equation (Equation 4) and $S_{edge}\left(e_{i},v_{j}\right)$ in Equation (Equation 5) are similarities of neighboring nodes and edges, respectively, and λ and δ in Equation (Equation 6) are their weight coefficients.

In Equation (Equation 4), ${\left[a\right]}_{+}$ denotes the positive part of *a*, while ϵ>0 signifies the maximum admissible distance among equivalent entities, and its value can be specified as the maximum embedding distance observed within the training set for equivalent entities. D(x,y) denotes the embedding distance between entities *x* and *y*. The computation of $S_{node}\left(e_{i},v_{j}\right)$ is based on the concept that the similarity between $e_{i}$ and $v_{j}$ increases as the number of equivalent entities in $V_{e_{i}}$ and $V_{v_{j}}$ grows. To satisfy this condition, our approach involves identifying the node *y* in $V_{v_{j}}$ with the closest embedding distance $\min_{y\in V_{v_{j}}}D$ to *x*. If the distance $\min_{y\in V_{v_{j}}}D\left(x,y\right)$ is smaller than the threshold ϵ, we consider *y* as a potential equivalent node of *x*. In such cases, we accumulate the difference between ϵ and $\min_{y\in V_{v_{j}}}D\left(x,y\right)$ into $S_{node}\left(e_{i},v_{j}\right)$. Conversely, if the embedding distance exceeds ϵ, we discard *y* as a potential equivalent node. In essence, the higher the count of potential equivalent nodes in $V_{e_{i}}$ and $V_{v_{j}}$, and the smaller the embedding distance among these potential equivalents, the greater the theoretical validity of equivalent entities, resulting in a larger value of $S_{node}\left(e_{i},v_{j}\right)$.

In Equation (Equation 5), edge $\left(x_{j},y_{j}\right)$ satisfying the specified condition cond is regarded as analogous to $\left(x_{i},y_{i}\right)$. The term $\sum_{\left(x_{i},y_{i}\right)\in E_{e_{i}}}\left|\left\{\left(x_{j},y_{j}\right)|cond\right\}\right|$ computes the count of edges in $E_{v_{j}}$ that meet the condition cond, and thereby quantifies their similarity to edges in $E_{e_{i}}$. As the number of such similar edges in $E_{e_{i}}$ and $E_{v_{j}}$ increases, so does the value of $S_{node}\left(e_{i},v_{j}\right)$.

Equation (Equation 6) considers both node and edge similarities in the graph and can be utilized to obtain the similarity between two local structures. A higher value of $S(LS\left(e_{i}\right),LS\left(v_{j}\right))$ indicates greater similarity between the two local structures.

### 4.5. Rearrangement Stage

This part is the most crucial component of our proposed approach. In this step, we rearrange the nodes in the candidate set matrix obtained in the previous step. The rearrangement process is depicted in Algorithm 1 and Figure 3, which takes as input the entity embedding matrix vec generated by the embedding module, the testing data $\left(E_{s},E_{t}\right)$, the candidate set size cand, and the adjacency matrix adj that encodes the structural information of the KGs. The algorithm outputs the rearranged candidate sets $CAND_{new}$ and the corresponding values of Hits@1.

In extant methodologies, the alignment module predominantly relies on the distance metric as the criterion for exploring cross-lingual correspondences. In this study, we present a novel metric called **Joint Similarity** as delineated below:(7)$$D\left(e_{i},v_{j}\right)=\left|D\left(e_{i},v_{j}\right)-S\left(LS\left(e_{i}\right),LS\left(v_{j}\right)\right)\right|.$$
which stipulates that the proximity between two nodes ought to decrease concomitantly with the enhancement in the similarity of their corresponding local structures.

The details of the implementation of the algorithm are described as follows. Firstly, for each node $e_{i}$ in the testing data from the source KG, we construct a map with the entity node’s id of $e_{i}$ as the key and its sum of ranking as the value and save its local structure into a sparse matrix. When traversing each node $e_{j}$ in the candidate set of $e_{i}$, we record the ranking of $e_{j}$ at first, and then add it with local structural similarity. Here, the ranking is defined as the size of the candidate set minus the subscript of the node in the candidate set. Then, we can obtain a map of the candidate set for $e_{i}$. By rearranging this map by value from largest to smallest, we obtain a new candidate set while considering local structural similarity. In theory, the ground truth cross-lingual correspondence in the new candidate set is ranked higher than in the original candidate set. Since not all nodes in the original candidate set satisfy local structural similarity, the length of the new candidate set is reduced, making it difficult to compute values other than Hits@1. Our approach focuses on improving Hits@1, i.e., alignment accuracy improvement.

### 4.6. Complexity Analysis

Algorithm 1 presents the key methodology of our proposed algorithm. It consists of two nested loops and a conditional structure for judgment. The outer loop traverses every node $e_{i}$ in the source knowledge graph $KG_{1}$, with a time complexity of $O(n_{e_{s}})$, where $n_{e_{s}}$ denotes the total number of entities in $KG_{1}$. Meanwhile, the inner loop iterates through each candidate node $v_{j}$ in the candidate set of $e_{i}$, with a time complexity of O(cand). Within this loop, the algorithm acquires the local structure and computes the similarity of local structures with a time complexity of $O(n_{r_{t}})$, where $n_{r_{t}}$ represents the total number of edges in the target graph. Furthermore, if the local structure of $e_{i}$ is similar to that of $v_{j}$, the algorithm proceeds to compute their similarity. In summary, the overall time complexity of the algorithm is $O(cand\times n_{e_{s}}\times n_{r_{t}})$.

The adoption of a fixed value for cand, limited to the range of [1,100], serves to significantly reduce the time complexity of our proposed approach. Moreover, the alignment process exclusively utilizes nodes present in the testing data rather than traversing the entire knowledge graph. This selective strategy substantially reduces the overall count of nodes and edges, resulting in a significant decrease in the algorithm’s time complexity.

## 5. Experiment

The proposed method is evaluated on the DBP15k datasets, aiming to address the following research questions:**RQ1**: To what extent can the integration of SSR with the baseline model’s embedding module enhance the prediction accuracy of the entity alignment task?**RQ2**: What is the impact of varying the rearranging ranges on the overall performance of the model?**RQ3**: How do the parameters λ and δ in Equation (Equation 6) affect the performance of the model?

### 5.1. Datasets

In our experiments, we utilized the DBP15k datasets proposed by Sun et al. [15], (Cross-lingual Entity Alignment via Joint Attribute-Preserving Embedding in *International Semantic Web Conference*, Springer, Cham, 2017) This dataset comprises four language-specific KGs extracted from DBpedia for English (En), Chinese (Zh), French (Fr), and Japanese (Ja), with each KG containing approximately 65 k–106 k entities. Additionally, the dataset provides aligned entity sets for English and three other languages, with each set containing more than 15 k cross-lingual entity pairs. Statistical information about the datasets is shown in Table 2.

#### Structure Statistic

The degree of nodes is one of the most notable metrics that can provide insight into graph structure. Therefore, we conducted an analysis of the degree-related information across the three language versions of the DBP15k dataset, which we present in Figure 4 and Table 3.

The node degree distribution across the three language versions of the DBP15k dataset is presented in Figure 3. The horizontal axis indicates various degree of nodes, while the vertical axis represents the number of nodes with the corresponding degree. Our analysis reveals that the distribution trends of node degrees in the knowledge graphs of the three language versions are relatively consistent. The number of nodes tends to increase and then decrease with the rise in node degrees, and the majority of nodes exhibit degrees between 2 and 5.

In Table 3, we present the statistics of node degree types across the DBP15k training set in three distinct language versions. The dataset $DBP15k_{ZH\text{\_}EN}$ contains 81 distinct types of node degrees, and $DBP15k_{JA\text{\_}EN}$ features 80 distinct types of node degrees, exhibiting only marginal variation from the former. However, $DBP15k_{FR\text{\_}EN}$ displays 108 different types of node degrees, which is likely due to the greater linguistic similarity between French and English compared to Chinese or Japanese. This linguistic proximity results in a higher count of cross-lingual entity links between French and English in DBpedia. The discrepancies in the number of node degree types across the three datasets are likely to have a significant impact on the outcome of the subsequent experiments.

### 5.2. Experiment Settings

In light of the current state of entity alignment research, we observe that GCN-based methods have exhibited the highest performance on the DBP15k dataset and do not necessitate the incorporation of extra training data (https://paperswithcode.com/sota/entity-alignment-on-dbp15k-zh-en (accessed on 5 July 2023)). While PSR [34] and EMGCN [35] exhibit superior performance, they necessitate supplementary training data, which is unsuitable in our case. Therefore, we have chosen several representative GCN-based methods (without extra training data) as baseline techniques for our comparative evaluation. Additionally, we have included JAPE, a translation-based model that has achieved the highest performance among non-GCN-based methods, as a baseline for comparison.

During the experiments, we conducted a control test by selecting only the SE variants of each baseline approach since our rearrangement method does not require the use of attribute information in the knowledge graph.

The inter-language links in each dataset are divided according to the gold standard. In all control experimental groups, we use 70% of the inter-language links as the test data and 30% of them as the train data. The main focus of the experiments is on the enhancement in Hits@1, i.e., the prediction accuracy of the corresponding entities across languages.

Apart from the hyperparameters that are included in our approach for the prior-order model, such as the division ratio between the training and validation sets and the learning rate, the length of the candidate set to be rearranged and the weight of neighbor are crucial hyperparameters. In our experiments, we test three candidate set lengths: 5, 10, and 50 to determine the optimal value, meanwhile we set the weights as λ=0.25, δ=0.5.

### 5.3. Baselines Information

In our controlled experiments, we chose four primary baseline approaches, including a representative translation-based model, JAPE, and for GCN-based models, GCN-Align, HGCN-JE, RDGCN, and AliNet. As our approach is focused on local structural similarity and does not require the utilization of attribute-related information, we only compared our method with the SE variants of each baseline approach in our comparative experiments.

**JAPE** [15] is a joint attribute-preserving embedding model for cross-lingual entity alignment, which jointly embeds the structures of two KGs into a unified vector space and further refines it by leveraging attribute correlations in the KGs. In our experiment, we set its hyperparameters to the best-performing values: d=75,α=0.1,β=0.05,δ=0.05. The learning rate of SE is set to 0.01 based on empirical evaluation.

**GCN-Align** [31] is a graph convolutional network-based approach that leverages entity relations to construct the network structure of GCNs, enabling it to generate embeddings for each entity in the source and target KGs. For our experiments, we selected the hyperparameters that have shown the most promising results, including setting $d_{s}=1000$ and $d_{a}=100$ for the source and target entity embeddings, respectively. Additionally, we set the margin $\gamma_{s}=\gamma_{a}=3$ in the loss function and empirically set β in the distance measure to 0.9.

**HGCN-JE** [6] is a joint entity–relation learning framework for entity alignment that minimizes human involvement and associated costs in seed alignment construction. For the experiment, we set the hyperparameters to γ=1, β=20, and the learning rate of 0.01. We also sample K=125 negative pairs every epoch.

**RDGCN** [18] is a novel relation-aware dual-graph convolutional network to incorporate relation information via attentive interactions between the knowledge graph and its dual relation counterpart, and further capture neighboring structures to learn better entity representations. In our experiments, we set $\beta_{1}=0.1$, $\beta_{2}=0.3$, and γ=1.0 as the hyperparameters. The hidden representations in the dual and primal attention layers have dimensions of d=300, $d^{\prime }=600$, and $\tilde{d}=300$. The dimension of hidden representations in GCN layers is 300. The learning rate is set to 0.001, and we sample K=125 negative pairs every 10 epochs.

**AliNet** [36] introduces distant neighbors to expand the overlap between their neighborhood structures, which employs an attention mechanism to highlight helpful distant neighbors and reduce noise. Subsequently, it controls the aggregation of both direct and distant neighborhood information using a gating mechanism. In our experiments, we set the number of layers of AliNet to 2, the activation function for neighborhood aggregation to tanh(), the one for the gating mechanism is RelU(), and the dimensions of the three layers (including the input layer) to 500, 400, and 300, respectively. For each pre-aligned entity pair, we sample 10 negative samples.

### 5.4. Results

Table 4, Table 5 and Table 6 show the results of all experiments on the DBP15k dataset.

The baseline methods exhibit distinct model characteristics, and can be categorized into two types: node-embedding models and structure-embedding models. Specifically, node-embedding models such as JAPE and GCN-Align solely consider node-level features, while structure-embedding models such as AliNet, HGCN-JE, and RDGCN learn not only node features, but also consider the surrounding environment information of the nodes.

In this study, we conducted a controlled experiment to evaluate the structural embedding variants (SEs) of all of the baseline methods. We computed their Hits@k values with k set to 1, 5, and 10. Subsequently, we employed the proposed SSR method to rearrange the corresponding candidate sets of cross-lingual entities obtained for each model with the corresponding model SSR variants. Here, XXX–SSR–c denotes the model utilizing an embedding module named XXX in combination with SSR, with the candidate set rearranging range set to cand=c.

As the SSR method only requires Hits@1 evaluation metrics, we report only the results for the Hits@1 column for the SSR variants. For the controlled experiments of each SSR variant with the original model, we emphasize the optimal Hits@1 results in **bold**, the suboptimal Hits@1 results underlined, and annotate the optimal results on the entire data set with * in the upper right corner of the values.

#### 5.4.1. Rearranging Range

As mentioned earlier, an important parameter of the algorithm proposed in this paper is the length of the rearrangement range of the candidate set, and this parameter has an impact on the performance of the model, which is reflected in the experimental results of different SSR variants. From the experimental results, we can tentatively conclude that the prediction accuracy Hits@1 of the variant model decreases as the value of cand increases when the value of cand is taken in the range of three numbers 5,10,50.

We apply the SSR method to the JAPE and GCN-Align models, and the experimental results show that the SSR variants of Hits@1 have a higher improvement relative to the original model when cand is taken as 5 and 10. Even when cand is taken to be 50, the SSR variants achieve a boost. This is due to the fact that both the JAPE and GCN-Align models are simpler node-embedding models, and the quality of the learned entity embeddings is average. The local structure-based rearrangement is able to adjust the distribution of nodes in the candidate set, thus achieving higher prediction accuracy.

For the AliNet method, Hits@1 improves to different degrees when the cand of the variant is taken as 5 and 10, but the variant with cand=50 does not perform as well as the original model. Furthermore, when we apply SSR to the HGCN-JE and RDGCN embedding modules, Hits@1 improves slightly only when cand takes 5. This is because the neural network designs of AliNet, HGCN-JE, and RDGCN are more complex. Among them, HGCN-JE introduces highway GCN, while RDGCN introduces relation-aware dual-graph. Although the training dataset of these two methods is limited to DBP15k, they can generate more data based on DBP15k due to the characteristics of the network structure, which is essentially equivalent to using exogenous data. AliNet introduces “distant neighbors” and an attention mechanism to consider the environmental information around the nodes, which also helps to improve the quality of the entity embedding vectors.

Through experimental observation, we have discovered a significant correlation between the lift rate of Hits@5 of the original model relative to Hits@1 of the original model, and the lift rate of Hits@1 of the SSR variant with cand=5 relative to Hits@1 of the original model. We conducted an in-depth analysis and comparison of these relationships. The results are presented in Figure 5. The horizontal axis of each figure indicates the original lift rate, which is the lift rate of Hits@5 of the original model relative to Hits@1 of the original model. The vertical axis represents the model lift rate, which is the lift rate of Hits@1 of the SSR variant with cand=5 relative to Hits@1 of the original model. Our analysis reveals a clear positive correlation between the two, where an increase in the original lift rate corresponds to an increase in the model lift rate.

#### 5.4.2. SSR vs. JAPE

In our comparative experiments, we investigate the effectiveness of the structure-enhanced rearrangement method by applying it to JAPE and evaluating its impact on entity alignment accuracy across three different cross-lingual KG datasets: $DBP15k_{ZH\text{\_}EN}$, $DBP15k_{JA\text{\_}EN}$, and $DBP15k_{FR\text{\_}EN}$. Our results indicate that the performance improvement of JAPE varies across different datasets, with the largest improvement achieved on $DBP15k_{ZH\text{\_}EN}$, followed by $DBP15k_{JA\text{\_}EN}$, and the smallest improvement on $DBP15k_{FR\text{\_}EN}$. Specifically, the accuracy rate of Hits@1 can be increased by around 5% when cand is set to 5, and still improved by about 1% even when cand is increased to 50. Such improvements are statistically significant and demonstrate the effectiveness of our approach.

The structure embedding part of JAPE uses a traditional translation-based model, i.e., given a relational triple (h,r,t), we expect h+r≈t. Such a simple model neglects the impact of the local structural information on the embeddings of entities, thereby presenting an opportunity to enhance JAPE’s performance through our proposed structure-enhanced rearrangement approach.

#### 5.4.3. SSR vs. GCNs

In the comparison experiments with GCNs, we observed that the original GCNs approach yielded almost identical Hits@1 results across three distinct datasets. On the other hand, our proposed method showed improvements on all three datasets, with a 5% improvement on $DBP15k_{ZH\text{\_}EN}$, 4.5% improvement on $DBP15k_{JA\text{\_}EN}$, and 3.6% on the $DBP15k_{FR\text{\_}EN}$ dataset.

The GCNs method trains by optimizing the loss function globally for all nodes in the graph. Our method applies a structure-enhanced rearrangement to the candidate set, which takes into account the local structural similarity of the graph. Unlike the GCN model, which only considers the distance similarity between single nodes in the alignment module, our approach also considers the similarity of nodes and their surrounding structures when evaluating equivalent entities, resulting in higher final values of Hits@1.

#### 5.4.4. SSR vs. AliNet

We conducted experiments on the DBP15k dataset using the AliNet model. Notably, our findings indicate that AliNet achieved comparable results across all three versions, without exhibiting the significant performance gains observed in the HGCN and RDGCN models on $DBP15k_{FR\text{\_}EN}$. Moreover, a careful examination of the experimental results reveals that the integration of our proposed approach into the AliNet model led to a noteworthy improvement of 2% to 3% on each of the three datasets.

#### 5.4.5. SSR vs. HGCN

The performance of HGCN on the DBP15k benchmark dataset exhibits significant improvement compared to JAPE and GCNs. In particular, HGCN achieves the highest accuracy on the $DBP15k_{FR\text{\_}EN}$ dataset, where JAPE and GCNs show inferior performance.

As previously noted, there are more structural variations present in the $DBP15k_{FR\text{\_}EN}$ dataset compared to $DBP15k_{ZH\text{\_}EN}$ and $DBP15k_{JA\text{\_}EN}$. This may explain why HGCN achieves a higher accuracy rate on the $DBP15k_{FR\text{\_}EN}$ dataset.

Upon applying our proposed approach to the HGCN model, we observed an improvement of over 1% on both the $DBP15k_{ZH\text{\_}EN}$ and $DBP15k_{JA\text{\_}EN}$ datasets, but only a 0.7% improvement on $DBP15k_{FR\text{\_}EN}$.

#### 5.4.6. SSR vs. RDGCN

In alignment with HGCN, the performance of RDGCN was evaluated across three distinct versions of the dataset. Notably, on the $DBP15k_{FR\text{\_}EN}$, RDGCN exhibited superior performance relative to the other datasets and achieved higher values of the Hits@1 metric when compared with HGCN. This discernible advantage may be attributed to the incorporation of a dual-relation knowledge graph by RDGCN, which effectively harnesses the relational intelligence embedded within the knowledge graph.

By employing the RDGCN embedding module in conjunction with our proposed methodology, our approach has yielded an increase of approximately 1% in the $DBP15k_{ZH\text{\_}EN}$ and $DBP15k_{JA\text{\_}EN}$ datasets. However, in the $DBP15k_{FR\text{\_}EN}$ dataset, the improvement is only 0.27%.

#### 5.4.7. Case Study

Figure 6 showcase the results of the experiments of both the baseline models and their respective SSR variants on the $DBP15k_{ZH\text{\_}EN}$ dataset. Each figure presents a horizontal axis denoting different types of node degrees and a vertical axis representing the number of entities. The blue squares depict the number of nodes that exhibit an increase in ranking for the actual cross-lingual equivalent entity within the candidate set upon rearrangement, denoted as “hit”. On the other hand, the orange circles represent the total number of declines, referred to as “miss”. Essentially, the blue squares indicate a positive impact of rearrangement, while the orange circles signify a negative effect. Consequently, the disparity between these two lines highlights the improvement effect of the SSR variant concerning the original model at different node degrees, where a larger difference denotes a more favorable improvement effect.

Upon careful examination of Figure 6, it becomes evident that the distribution of node degrees adheres to the statistical information of the dataset, as demonstrated in Figure 4.

Initially, we conducted a thorough analysis of graphs Figure 6. These graphs revealed that regardless of the degree of the node, ranging from 0 to 19, the total number of genuine cross-lingual equivalent entities in the candidate set that are ranked up after rearrangement consistently exceeded the total number of entities ranked down. Additionally, the difference between the number of hits and misses was always maintained at a higher value, with a minimum difference of 40. This phenomenon is the key factor responsible for the significant enhancement in the performance our method in comparison to the JAPE and GCN-Align embedding models.

Upon examining Figure 6, we observed that the disparity between the number of “hits” and “misses” was not substantial (the highest difference did not surpass 40) even in regions with the most densely distributed nodes. Notably, for the SSR variant of HGCN, there was only a significant difference in nodes with degrees of 2 and 3. Furthermore, as the node degree increased, the “miss” lines tended to exceed the “hit” lines.

To sum up, our proposed method has effectively enhanced the prediction outcomes for nodes with low degrees. However, the performance of the variant model shows a gradual deterioration for nodes with higher degrees.

#### 5.4.8. Ablation Experiments and Sensitivity Analysis

In this paper, we propose a novel approach for computing the local structural similarity of nodes in a graph. Our method takes into account both the neighboring node information and the neighboring edge information present in the node’s surrounding environment, which are denoted by the node information weight λ and the edge information weight δ in Equation (Equation 6), respectively. To demonstrate the significance of these two factors, we conduct ablation experiments and a sensitivity analysis in this section.

We present two sets of experiments to investigate the significance of neighboring node and edge information in our proposed approach. Specifically, we conducted the first set of experiments by maintaining δ=0, while varying the value of node weights λ to determine the effect of neighboring node information. The second set of experiments involved fixing the node weights at λ=0 and modifying the value of edge weights δ to examine the impact of neighboring edge information.

First, we investigate the significance of neighboring node information in the local structure of candidate set nodes. To this end, we perform experiments on several embedding modules, including JAPE, GCN-Align, AliNet, HGCN-JE, and RDGCN, by fixing the edge weight coefficient at δ=0 and adjusting the value of λ in the SSR variants. Each model is trained for 200 iterations, and the Hits@1 results are recorded and presented in Table 7, Table 8, Table 9, Table 10 and Table 11.

The values of λ in the SSR variant of JAPE have been fine-tuned and the experimental outcomes are presented in Table 7. In the course of progressively augmenting the node weight λ from 0 to 5.0, the corresponding values of Hits@1 of the variant model exhibit a predominantly declining trend across all datasets. Optimal performance, as measured by Hits@1, is achieved when λ is set to 1 on the $DBP15k_{ZH\text{\_}EN}$ and $DBP15k_{JA\text{\_}EN}$ datasets. This finding indicates that in the local structure of nodes, the information emanating from neighboring nodes plays an equally crucial role as the embedding distance.

The experimental results of applying SSR on top of the GCN-Align embedding module are shown in Table 8. Similar to JAPE, the values of Hits@1 of the variant model on each dataset generally exhibit a decreasing trend as the node information weight λ is increased from 0 to 5.0, with the highest value achieved at λ=1.0 only on the $DBP15k_{JA\text{\_}EN}$ dataset. These findings suggest that neighboring node information, in addition to the distance between node embeddings, plays a crucial role in the local structure of nodes when rearranging the nodes in the candidate set obtained from the original GCN-Align model, and such information significantly affects the performance of the final model.

The experimental outcomes of applying SSR on top of AliNet are presented in Table 9. As the node information weight λ is gradually increased from 0 to 5.0, the variant model exhibits a consistent decreasing trend in the Hits@1 performance metric across all datasets. The best performance is achieved at λ=1.0 for all datasets. This finding underscores the importance of incorporating neighboring node information in the local structure of nodes for the rearrangement of nodes in the candidate set generated by the original AliNet model.

The three sets of control experiments described above involved a uniform selection of λ values ranging from 0 to 5.0. Node embedding models such as JAPE and GCN-Align generate subpar embeddings for cross-lingual entity alignment and exhibit poor performance in this task. However, with the addition of SSR, the variant models are able to leverage both vector distance and node local structure similarity when searching for cross-lingual counterpart entities. Notably, the information of neighboring nodes in the node local structure plays an equally important role as the embedding vector distance. In contrast, AliNet, a structure-embedding model, considers the information of neighbors at a greater distance from the nodes. As such, the information of neighboring nodes in the local structure surrounding nodes can substantially improve the performance of the variant model when combined with SSR.

The experimental results of combining SSR with the embedding module of HGCN-JE are presented in Table 10. As the node information weight coefficient λ is progressively increased from 0 to 0.50, the performance of the variant model exhibits an initial increase followed by a decrease, and the best performance is obtained when λ is set to 0.25, 0.25, and 0.10 on each dataset. These findings suggest that, although the original HGCN-JE model performs well, incorporating neighboring node information in the local structure of the nodes can further enhance the predictive accuracy of the variant model.

When applying our method to the original RDGCN model, as shown in Table 11, the node information weight coefficient λ is gradually increased from 0 to 0.50, resulting in a similar trend to HJ–SSR–5, where the variant model performance first increases and then decreases. The best results are achieved at λ=0.25, λ=0.25, and λ=0.10 on each language version dataset, respectively. These experimental results demonstrate the critical role that neighboring node information in the local structure plays in positively influencing the prediction results and improving the overall performance of the model.

In comparison to the three control experiments mentioned earlier, the traversal interval of λ in our study is narrower. This is attributed to the competence of the two models which excel in learning high-quality knowledge graph embeddings. The prediction accuracy of cross-lingual corresponding entities, solely reliant on distance similarity, is already remarkably high. Consequently, the impact of neighboring nodes’ information weights diminishes in comparison to the first three node-level models. Nonetheless, our proposed SSR variant manages to achieve discernible improvements over the original model.

The aforementioned experiments lead to the conclusion that the node information plays a crucial role in the model, significantly impacting its performance sensitivity to variations in λ.

After verifying the significance of the local structure node information, we proceeded to investigate the role of edge information. In this set of experiments, we set the node weight coefficient λ to 0 and uniformly sampled edge weight coefficients in the range [0,1.0]. The experimental results of the variant model are presented in Table 12, Table 13, Table 14, Table 15 and Table 16. Since the Hits@1 results of the model on each dataset remain almost constant when varying the value of δ, we combine the results of different δ values.

The empirical findings indicate that the incorporation of neighboring edge information within the local structure makes a negligible contribution to the models’ performance, with this parameter δ demonstrating low sensitivity.

## 6. Conclusions

In this paper, we propose a novel local structure-based rearrangement method that exploits the structural similarity between the source and target knowledge graphs and considers not only the distance between entity embeddings in the alignment phase, but also incorporates the feature of the local structure constituted by the entities and their surrounding nodes into the evaluation metric of entity similarity in order to improve the alignment accuracy, which is reflected in the rearrangement of the candidate set nodes. Experiments on three cross-lingual datasets of DBP15k demonstrate that the proposed approach can achieve promising performance. Through a comprehensive case study, ablation experiments, and a sensitivity analysis, we reveal that the information derived from neighboring nodes within the entity’s local structure exerts a more substantial influence on the model and exhibits heightened sensitivity. Notably, our method demonstrates superior capabilities in handling nodes with smaller degrees. However, for nodes characterized by larger degrees and intricate structures, the improvement effect after rearrangement becomes less pronounced. This aspect poses a pivotal challenge, impeding the further enhancement in our method’s performance.

## Figures and Tables

**Figure 1 sensors-23-07096-f001:**
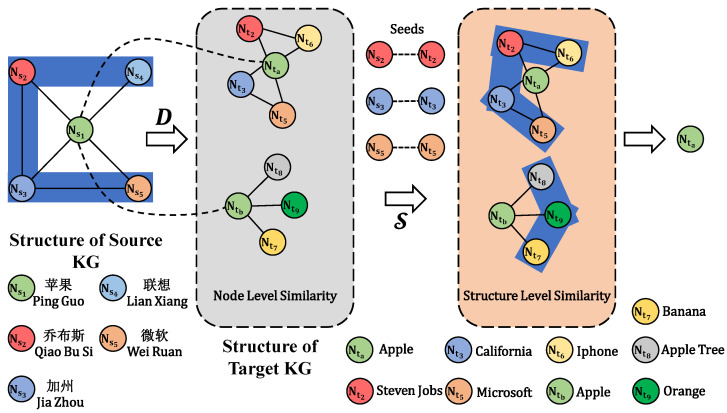
An example of cross-lingual KG entity alignment.

**Figure 2 sensors-23-07096-f002:**
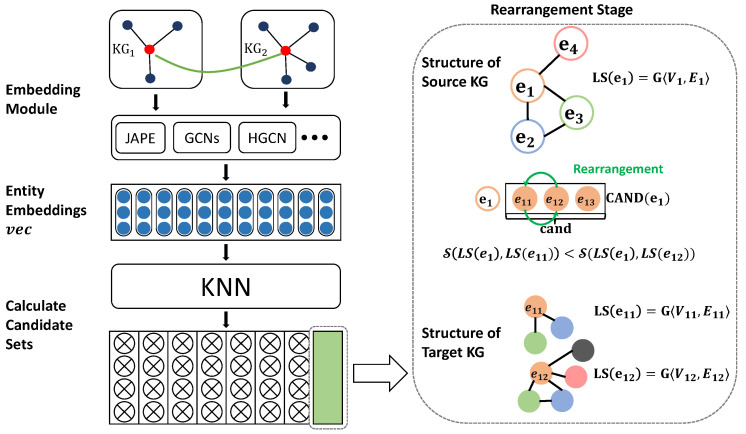
Overall architecture of the proposed approach.

**Figure 3 sensors-23-07096-f003:**
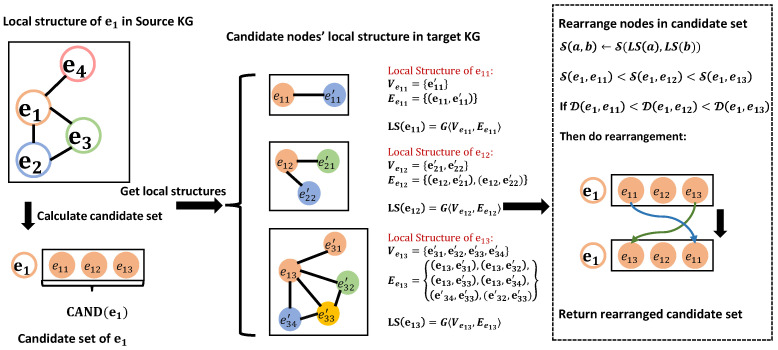
Entity rearrangement process. Firstly, we calculate the local structure of all nodes within the candidate set of $e_{1}$ via Equation (Equation 3). Secondly, we compute the local structural similarity between $e_{1}$ and the nodes within the candidate set using Equation (Equation 6). Following this, the joint similarities are determined through Equation (Equation 7). Lastly, to complete the rearrangement, the nodes within the candidate set are rearranged based on their respective values of joint similarity.

**Figure 4 sensors-23-07096-f004:**
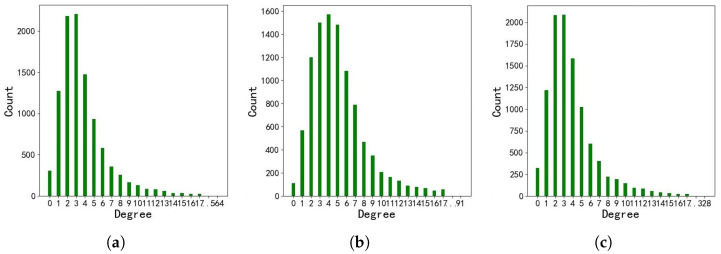
Degree distribution of local structure in DBP15k. (**a**) DBP15k${}_{ZH\text{\_}EN}$; (**b**) DBP15k${}_{JA\text{\_}EN}$; (**c**) DBP15k${}_{FR\text{\_}EN}$.

**Figure 5 sensors-23-07096-f005:**
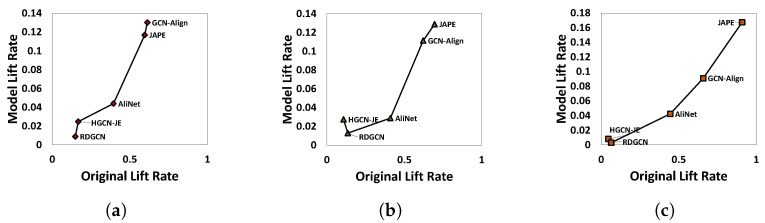
Models’ performance on DBP15k. (**a**) DBP15k${}_{ZH\text{\_}EN}$; (**b**) DBP15k${}_{JA\text{\_}EN}$; (**c**) DBP15k${}_{FR\text{\_}EN}$.

**Figure 6 sensors-23-07096-f006:**
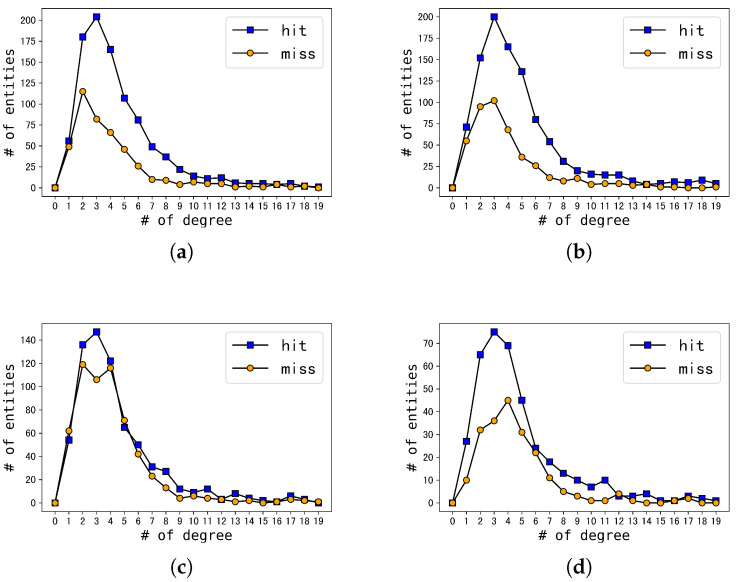
Influence via SSR. (**a**) SSR vs. GCNs; (**b**) SSR vs. JAPE; (**c**) SSR vs. HGCN; (**d**) SSR vs. RDGCN.

**Table 1 sensors-23-07096-t001:** Important notation.

Symbol	Definition
KG=(E,R,T)	A KG consisting of sets of entities *E*, relations *R*, and triples *T*.
$E_{s}$, $E_{t}$	The sets of entities from $KG_{1}$ and $KG_{2}$.
$e_{i}$	An entity from KG.
vec	The entity embeddings.
$CAND(e_{i})$	The candidate set of entity $e_{i}$.
$CAND(E_{s})$	The candidate sets of $E_{s}$.
$CAND_{new}\left(e_{i}\right)$	The rearranged candidate set of entity $e_{i}$.
$CAND_{new}$	The rearranged candidate sets of $E_{s}$.
cand	The size of candidate set.
$rank(e_{i})$	The ranking of entity $e_{i}$.
G=〈V,E〉	A graph *G* consisting of set of nodes *V* and set of edges *E*.
S	The similarity of local structures.
index(list,e)	The index of *e* in list.

**Table 2 sensors-23-07096-t002:** Statistical information of DBP15k.

Dataset	Languages	Entities	Relations	Attributes	Rel. Triples	Attr. Triples
ZH_EN	Chinese	66,469	2830	8113	153,929	379,684
	English	98,125	2317	7173	237,674	567,755
JA_EN	Japanese	65,744	2043	5882	164,373	354,619
	English	95,680	2096	6066	233,319	497,230
FR_EN	French	66,857	1379	4547	192,191	528,665
	English	105,889	2209	6422	278,590	576,543

**Table 3 sensors-23-07096-t003:** Cardinality of local structure diversity.

Dataset	$DBP15k_{ZH\text{\_}EN}$	$DBP15k_{JA\text{\_}EN}$	$DBP15k_{FR\text{\_}EN}$
**Number**	81	80	108

**Table 4 sensors-23-07096-t004:** Experiment results on DBP15k${}_{ZH\text{\_}EN}$.

Type	Model	Hits@1	Hits@5	Hits@10
NEM ^1^	JAPE	39.39	62.86	71.02
	JAPE–SSR–5	**44.00**	-	-
	JAPE–SSR–10	43.91	-	-
	JAPE–SSR–50	40.65	-	-
NEM	GCN-Align	39.16	63.16	69.93
	GCNs–SSR–5	**44.26**	-	-
	GCNs–SSR–10	43.99	-	-
	GCNs–SSR–50	41.32	-	-
SEM ^2^	AliNet	51.71	72.13	85.07
	AlN–SSR–5	**53.98**	-	-
	AlN–SSR–10	53.39	-	-
	AlN–SSR–50	49.76	-	-
SEM	HGCN-JE	69.31	80.92	84.70
	HJ–SSR–5	**71.02**	-	-
	HJ–SSR–10	68.30	-	-
	HJ–SSR–50	59.98	-	-
SEM	RDGCN	72.02	82.70	85.58
	RG–SSR–5	**72.66** *	-	-
	RG–SSR–10	67.26	-	-
	RG–SSR–50	62.90	-	-

^1^ Abbreviation of node-embedding model. ^2^ Abbreviation of structure-embedding model.

**Table 5 sensors-23-07096-t005:** Experiment Results on DBP15k${}_{JA\text{\_}EN}$.

Type	Model	Hits@1	Hits@5	Hits@10
NEM	JAPE	33.84	57.39	67.03
	JAPE–SSR–5	**38.20**	-	-
	JAPE–SSR–10	38.15	-	-
	JAPE–SSR–50	35.53	-	-
NEM	GCN-Align	40.40	65.50	73.44
	GCNs–SSR–5	**44.90**	-	-
	GCNs–SSR–10	44.09	-	-
	GCNs–SSR–50	40.73	-	-
SEM	AliNet	51.70	72.88	79.62
	AlN–SSR–5	**53.19**	-	-
	AlN–SSR–10	52.30	-	-
	AlN–SSR–50	48.78	-	-
SEM	HGCN-JE	76.09	84.26	87.99
	HJ–SSR–5	**78.18**	-	-
	HJ–SSR–10	70.50	-	-
	HJ–SSR–50	61.93	-	-
SEM	RDGCN	77.84	88.30	90.79
	RG–SSR–5	**78.84** *	-	-
	RG–SSR–10	71.40	-	-
	RG–SSR–50	62.93	-	-

**Table 6 sensors-23-07096-t006:** Experiment Results on DBP15k${}_{FR\text{\_}EN}$.

Type	Model	Hits@1	Hits@5	Hits@10
NEM	JAPE	28.75	54.88	64.43
	JAPE–SSR–5	**33.56**	-	-
	JAPE–SSR–10	33.45	-	-
	JAPE–SSR–50	31.19	-	-
NEM	GCN-Align	40.11	66.51	75.98
	GCNs–SSR–5	**43.76**	-	-
	GCNs–SSR–10	42.83	-	-
	GCNs–SSR–50	38.60	-	-
SEM	AliNet	52.35	75.69	82.40
	AlN–SSR–5	**54.57**	-	-
	AlN–SSR–10	53.36	-	-
	AlN–SSR–50	48.71	-	-
SEM	HGCN-JE	88.95	93.12	94.75
	HJ–SSR–5	**89.67** *	-	-
	HJ–SSR–10	77.54	-	-
	HJ–SSR–50	64.29	-	-
SEM	RDGCN	88.79	94.57	95.91
	RG–SSR–5	**89.06**	-	-
	RG–SSR–10	76.25	-	-
	RG–SSR–50	64.40	-	-

**Table 7 sensors-23-07096-t007:** Node weight for JAPE–SSR–5.

λ	$\mathrm{DBP}15k_{\mathrm{ZH}\text{\_}\mathrm{EN}}$ *	$\mathrm{DBP}15k_{\mathrm{JA}\text{\_}\mathrm{EN}}$	$\mathrm{DBP}15k_{\mathrm{FR}\text{\_}\mathrm{EN}}$
0	39.39	33.84	28.75
0.5	43.72	36.10	**32.44**
1.0	**44.24**	**36.22**	32.11
1.5	43.54	35.54	31.47
2.0	43.42	35.23	31.34
2.5	42.90	34.77	30.95
3.0	42.90	34.77	30.95
3.5	42.59	34.44	30.84
4.0	42.59	34.44	30.84
4.5	42.27	34.30	30.68
5.0	42.27	34.30	30.38

* Hits@1 on DBP15k${}_{ZH\text{\_}EN}$.

**Table 8 sensors-23-07096-t008:** Node weight for GCNs–SSR–5.

λ	$\mathrm{DBP}15k_{\mathrm{ZH}\text{\_}\mathrm{EN}}$	$\mathrm{DBP}15k_{\mathrm{JA}\text{\_}\mathrm{EN}}$	$\mathrm{DBP}15k_{\mathrm{FR}\text{\_}\mathrm{EN}}$
0	39.16	40.40	40.11
0.5	**43.63**	43.63	**42.41**
1.0	43.56	**44.15**	42.26
1.5	42.76	43.44	41.41
2.0	42.53	43.23	41.33
2.5	42.15	42.71	40.74
3.0	42.15	42.71	40.74
3.5	41.83	42.40	40.56
4.0	41.83	42.40	40.56
4.5	41.63	42.20	40.19
5.0	41.63	42.20	40.19

**Table 9 sensors-23-07096-t009:** Node weight for AlN–SSR–5.

λ	$\mathrm{DBP}15k_{\mathrm{ZH}\text{\_}\mathrm{EN}}$	$\mathrm{DBP}15k_{\mathrm{JA}\text{\_}\mathrm{EN}}$	$\mathrm{DBP}15k_{\mathrm{FR}\text{\_}\mathrm{EN}}$
0	51.71	51.70	52.35
0.5	**53.86**	**53.19**	**54.57**
1.0	53.75	52.96	53.81
1.5	53.05	52.23	52.47
2.0	52.85	51.88	52.20
2.5	52.27	51.36	51.79
3.0	52.27	51.36	51.79
3.5	51.70	51.04	51.30
4.0	51.70	51.04	51.30
4.5	51.46	50.72	51.05
5.0	51.46	50.72	51.05

**Table 10 sensors-23-07096-t010:** Node weight for HJ–SSR–5.

λ	$\mathrm{DBP}15k_{\mathrm{ZH}\text{\_}\mathrm{EN}}$	$\mathrm{DBP}15k_{\mathrm{JA}\text{\_}\mathrm{EN}}$	$\mathrm{DBP}15k_{\mathrm{FR}\text{\_}\mathrm{EN}}$
0	69.31	76.09	88.95
0.05	69.68	76.52	89.35
0.10	70.30	77.03	**89.67**
0.15	70.76	77.74	89.39
0.20	70.82	77.84	89.35
0.25	**71.02**	**78.18**	88.79
0.30	70.52	77.69	88.17
0.35	70.08	77.35	87.54
0.40	70.05	77.30	87.44
0.45	69.89	77.10	86.97
0.50	69.89	77.10	86.97

**Table 11 sensors-23-07096-t011:** Node weight for RG–SSR–5.

λ	$\mathrm{DBP}15k_{\mathrm{ZH}\text{\_}\mathrm{EN}}$	$\mathrm{DBP}15k_{\mathrm{JA}\text{\_}\mathrm{EN}}$	$\mathrm{DBP}15k_{\mathrm{FR}\text{\_}\mathrm{EN}}$
0	70.77	77.33	88.81
0.05	71.04	77.47	88.90
0.10	71.76	78.09	**89.06**
0.15	72.08	78.26	88.63
0.20	72.27	78.41	88.29
0.25	**72.53**	**78.54**	87.50
0.30	71.97	77.65	86.31
0.35	71.70	77.17	85.35
0.40	71.62	77.19	85.25
0.45	71.71	77.10	84.67
0.50	71.71	77.10	84.67

**Table 12 sensors-23-07096-t012:** Edge weight for GCNs–SSR–5.

δ	$\mathrm{DBP}15k_{\mathrm{ZH}\text{\_}\mathrm{EN}}$ *	$\mathrm{DBP}15k_{\mathrm{JA}\text{\_}\mathrm{EN}}$	$\mathrm{DBP}15k_{\mathrm{FR}\text{\_}\mathrm{EN}}$
0	39.16	40.40	40.11
[0.1,1.0]	39.16	40.40	40.11

* Hits@1 on DBP15k${}_{ZH\text{\_}EN}$.

**Table 13 sensors-23-07096-t013:** Edge weight for AlN–SSR–5.

δ	$\mathrm{DBP}15k_{\mathrm{ZH}\text{\_}\mathrm{EN}}$	$\mathrm{DBP}15k_{\mathrm{JA}\text{\_}\mathrm{EN}}$	$\mathrm{DBP}15k_{\mathrm{FR}\text{\_}\mathrm{EN}}$
0	51.71	51.70	52.35
[0.1,1.0]	51.57	51.70	52.35

**Table 14 sensors-23-07096-t014:** Edge weight for JAPE–SSR–5.

δ	$\mathrm{DBP}15k_{\mathrm{ZH}\text{\_}\mathrm{EN}}$	$\mathrm{DBP}15k_{\mathrm{JA}\text{\_}\mathrm{EN}}$	$\mathrm{DBP}15k_{\mathrm{FR}\text{\_}\mathrm{EN}}$
0	39.39	33.84	28.75
[0.1,1.0]	39.39	33.84	28.75

**Table 15 sensors-23-07096-t015:** Edge weight for RDGCN–SSR–5.

δ	$\mathrm{DBP}15k_{\mathrm{ZH}\text{\_}\mathrm{EN}}$	$\mathrm{DBP}15k_{\mathrm{JA}\text{\_}\mathrm{EN}}$	$\mathrm{DBP}15k_{\mathrm{FR}\text{\_}\mathrm{EN}}$
0	72.02	77.84	88.79
[0.1,1.0]	72.02	77.84	88.79

**Table 16 sensors-23-07096-t016:** Edge weight for HJ–SSR–5.

δ	$\mathrm{DBP}15k_{\mathrm{ZH}\text{\_}\mathrm{EN}}$	$\mathrm{DBP}15k_{\mathrm{JA}\text{\_}\mathrm{EN}}$	$\mathrm{DBP}15k_{\mathrm{FR}\text{\_}\mathrm{EN}}$
0	69.31	76.09	88.95
[0.1,1.0]	69.31	76.09	88.95

## Data Availability

The data in experiments is available on the Internet and can be accessed easily.

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
