# Peer review of "Enhancing Cross-Lingual Entity Alignment in Knowledge Graphs through Structure Similarity Rearrangement"

_sensors, 2023, doi:10.3390/s23167096_

Round 1

Reviewer 1 Report

In the reviewer’s point of view, this paper’s main contribution is that the authors have proposed a novel alignment model called SSR, which leverages the node embedding algorithm in graphs to select candidate entities and then rearranges them by local structural similarity in the source and target knowledge graphs. Particularly, they have also demonstrated the effectiveness of our approach on the DBP15k dataset, showing that it outperforms existing methods while requiring less time. The results are quite interesting to the community, and the manuscript is well written. Nevertheless, there are still a few issues that need to be resolved before possible final publication, which are listed in the following:

1.      At the bottom of Page 18 of 21, it is mentioned "However, our proposed method can still achieve a small but significant improvement in prediction accuracy, demonstrating the feasibility of our approach." For better understanding of the potential readers, it is highly recommended the authors might at least add one or two sentences in the following section, elucidating how the proposed SSR approach may further improve the prediction accuracy, and what might be the challenges preventing the further improvements.

2.      There are up to 5 equations throughout the manuscript, whose format seem somehow not uniform. For example, it is suggested that each equation ends with a period.

3.      Besides, it is also recommended that each caption for every plot in the manuscript shall end with a period. For example, the caption for Figure 5.

4.      The Figure 4 is on Page 10 of 21, but the content mentioning Figure 4 is on Page 16 of 21. In other words, the authors mentioned Figure 5 and Figure 6 before mentioning Figure 4, which might cause confusion and needs to be adjusted accordingly.

5.      In addition, Figure 4 has 3 sub-plots, so it is recommended that those sub-plots are titled Figure 4(a), 4(b) and 4(c) to avoid confusion.

6.      Last but not least, please carefully edit your manuscript by correcting typos, and please have a person fluent in English proofread your paper to address all the language issues.

If English is your second language and editing services are required for your manuscript, you may find the following editing websites useful for a cost:

1. IEEE Professional Editing Services at http://www.prof-editing.com/ieee/

2. American Journal Experts at http://www.journalexperts.com/

Reviewer 2 Report

This paper proposes a rearrangement method based on graph structural similarity for cross-lingual entity alignment task. Compared to existing alignment algorithms that only consider node representations, this algorithm takes into account the local structural similarity of the graph, resulting in a significant improvement in alignment performance. The overall experiment in this article is complete and yields good results. However, there are some issues that need to be improved

(1) There are some errors in Formula (4), where delta is not presented in the equation. In addition, please clarify that what is minimized in the first term on the right side of the equation.

(2) Please take Figure 3 as an example, provide a brief description of the execution process of the proposed algorithm.

(3) AliNet lacks the corresponding references.

(4) The ablation experiment is peculiar. In the formula (4), there are two parts need to be tested. Setting the contribution of one of them to 0 allows you to determine the contribution of that part in the algorithm. But in your experiment, you varying the lambda, is there some difference when you set different lambda values? According to the formula (4), all edge weights are the same, means that the edges share the same delta. How to vary the delta value? 

(5) Please label the corresponding nodes by e11 and e12 in Figure 2.

-

Reviewer 3 Report

This work proposes to exploit structural similarity in the problem of cross-lingual entity alignment.

The paper is presented well written and formatted, and easy to follow. It motivates appropriately the decisions behind the proposed approach. Its formalization in section 4 seems sound.

The experimental setup is principled, with well-known datasets and relevant baselines to compare to.

The experiments are conducted under the guide of relevant RQs, and their results are presented spanning across several tables and detailed analysis. The ablation studies in section 5.4.8 nicely complement the analysis and overall wrap up a significantly long discussion of the results.

Author Response

Thank you for your comments, which we will take as recognition and encouragement for our research and continue to explore in domain of EA (Entity Alignment) with a prudent and truthful attitude.

Round 2

Reviewer 2 Report

I have no other problems.